# Genome-Wide Identification and Characterization of Cysteine-Rich Receptor-Like Protein Kinase Genes in Tomato and Their Expression Profile in Response to Heat Stress

**Yahui Liu [1]**, **Zhengxiang Feng [2]**, **Weimin Zhu [1]**, **Junzhong Liu [2],*** and **Yingying Zhang [1],***

[1] Shanghai Key Laboratory of Protected Horticulture Technology, The Protected Horticulture Institute, Shanghai Academy of Agricultural Sciences, Shanghai 201403, China; liuyahui@saas.sh.cn (Y.L.); yy17@saas.sh.cn (W.Z.)

[2] State Key Laboratory of Conservation and Utilization of Bio-Resources in Yunnan and Center for Life Sciences, School of Life Sciences, Yunnan University, Kunming 650091, China; Fengzhengxiang11@163.com

* Correspondence: liujunzhong@ynu.edu.cn (J.L.); zhangyingying2018@saas.sh.cn (Y.Z.)

**Abstract:** During plant growth, development and stress adaption, receptor-like protein kinases (RLKs) are essential components in perceiving and integrating extracellular stimuli and transmitting the signals to activate the downstream signaling pathways. Cysteine-rich receptor-like protein kinases (CRKs) are a large subfamily of RLKs and their roles in modulating plant disease resistance are well elucidated. However, the roles of CRKs in plant abiotic stress responses, especially heat stress, are largely unknown. In this study, 35 *SlCRK* genes were identified in tomato (*Solanum lycopersicum*) based on the multiple sequence alignment and phylogenetic relationships. *SlCRK* genes are tandemly distributed on seven chromosomes and have similar exon–intron organization and common conserved motifs. Various phytohormone responsive, stress responsive cis-regulatory elements and heat shock elements are predicted in the promoter regions of *SlCRK* genes. Transcriptome analysis of tomato fruits under heat stress revealed that most *SlCRK* genes were downregulated upon heat treatment. GO enrichment analyses of genes that were co-expressed with *SlCRK* members have identified various stress responses related and proteasomal protein catabolic process related genes, which may be involved in heat stress signaling. Overall, our results provide valuable information for further research on the roles of *SlCRK*s in response to abiotic stress, especially heat stress.

**Keywords:** CRK; heat stress resistance; *Solanum lycopersicum*; genome-wide profiling; expression regulation



## 1. Introduction

Receptor-like protein kinases (RLKs) percept a variety of external and internal stimuli and transmit the input signal to induce the activated expression of appropriate target genes [1]. RLKs generally contain an amino-terminal signal sequence, an extracellular domain, a single transmembrane domain and a cytoplasmic domain with serine/threonine protein kinase activity [2]. RLKs can be divided into several sub-families, including leucine-rich repeat RLKs (LRR-RLKs), cysteine-rich repeat (CRR) RLKs (CRKs), domain of unknown function 26 (DUF26) RLKs, S-domain RLKs, and others [3]. CRKs contain two copies of the conserved C-X8-C-X2-C motif in their extracellular regions [4]. The role of DUF26 domain still remains elusive, while the cysteines have been proposed to form disulfide bridges as potential targets for thiol redox regulation and serve as sensors for reactive oxygen species (ROS), the common signaling molecules produced under various stresses [5]. The extracellular domain of CRKs perceives the extracellular ligands and transduces the signal to intracellular kinase domains. In *Arabidopsis thaliana*, 46 CRK members have been identified [5]. Many CRKs have been proven to play vital roles in biotic stress response. Over-expression of CRK5 or CRK13 enhances plant resistance

to *Pseudomonas syringae* [6,7] The signaling pathways mediated by several CRKs, such as BR-insensitive 1 (BRI1) [8] and FLAGELLIN-SENSITIVE2(FLS2) [9], have been well characterized in hormone perception and pathogen response. In addition, many CRKs are transcriptionally induced in response to abiotic stress conditions such as salicylic acid, ozone, UV light, drought and salt treatments [5,6,10–12]. Induced expression of the four structurally closely related CRKs, CRK4, CRK5, CRK19 and CRK20, leads to hypersensitive response-associated cell death in transgenic *Arabidopsis* [6,11]. CRK6 and CRK7 have been reported to mediate the responses to extracellular ROS production [13]. Two abiotic stress inducible CRK members, CRK36 and CRK45, interact with each other and negatively regulate ABA and osmotic stress signal transduction [14,15]. Besides, CRKs play vital roles in plant immunity. For example, TaCRK2 contributes to leaf rust resistance in wheat (*Triticum aestivum*) [16]. In cucumber (*Cucumis sativus*), a dominantly inherited powdery mildew resistance QTL, *Pm1.1*, contains two tandemly arrayed cysteine-rich receptor-like protein kinase genes [17].

Tomato (*Solanum lycopersicum*) is a widely cultivated and the most important vegetable worldwide. Tomato is a thermophilic horticultural crop, and sensitive to high temperature. Heat stress poses a serious threat to tomato fruit yield and quality [18–20]. Transcriptional profiling of maturing tomato microspores reveals the involvement of heat shock proteins, ROS scavengers, hormones and sugars in heat stress response [21]. SlMPK1, a mitogen-activated protein kinase, negatively regulates heat stress response by directly interacting with SlSPRH1, an antioxidant defense protein [22]. Overexpression of receptor-like kinase ERECTA improves thermotolerance in rice and tomato [23]. Transgenic tomato plants over-expressing betaine aldehyde dehydrogenase (BADH) and choline oxidase A (codA), key enzymes in betaine synthesis, exhibited heat tolerance, along with the elevated expression of heat stress responsive genes and the accumulation of HSP70 protein [24]. However, the regulatory network in response to heat stress in tomato is largely obscure, and the putative roles of *CRK* genes in tomato abiotic stress acclimation, especially heat stress, have not been investigated.

To uncover the potential roles of *SlCRK* genes in heat stress responses, we performed a genome-wide analysis of the tomato *SlCRK* gene family. Thirty-five putative *SlCRK* genes in tomato were identified and named according to chromosomal distributions. All *SlCRK* genes possessed various stress-responsive cis-elements, suggesting that *SlCRK* can respond to various environmental stimuli. Phylogenetic tree, gene structure and conserved motif analyses showed that *SlCRK* genes have conserved structure and motifs, and tandemly clustered *SlCRK*s are diverse. Transcriptome analyses of tomato fruits treated by heat stress revealed that *SlCRK* genes were mainly downregulated upon heat. GO enrichment analysis of co-expressed genes suggested that some stress responsive genes were significantly regulated. Our study has identified the tomato CPK gene family and provides important cues for further investigations on CRK-mediated network in plant abiotic stress responses.

## 2. Materials and Methods

### 2.1. Plant Materials and Growth Conditions

The tomato cultivar "Micro-Tom" is a model cultivar featured by small size, short life circle and capacity to grow under fluorescent lights at a high density. The seedlings were grown in pots inside a greenhouse maintained at 28 °C and 14/10 h (light/dark) photoperiod and 60–70% humidity. For heat stress treatment, four-week seedlings were treated at 42 °C for 0 h, 24 h, 48 h and 96 h, continuously. Leaves were collected at each point and frozen in liquid nitrogen immediately for total RNA extraction.

### 2.2. RNA Extraction and Expression Analysis

Total RNA was extracted from collected tomato fruit samples using the TRIzol reagent (Invitrogen, Carlsbad, CA, USA). The RNA concentration and purity of total RNA samples were checked using a Nanodrop2000 instrument (Thermo Fisher Scientific, Waltham, MA, USA) and electrophoresis in agarose gels. For mRNA-seq, the cDNA library was

constructed by Illumina TruseqTM RNA sample prep kit. After analysis of RIN values using an Agilent 2100 Bioanalyzer Lab-on-Chip system (Agilent, Santa Clara, CA, USA), the pair-end sequencing was performed on Illumina HiSeq X Ten platform.

### 2.3. Expression Analysis of SlCRK Genes and GO Enrichment of Co-Expressed Genes of SlCRK Genes

The mRNA-seq were conducted and analyzed by Oebiotech Corporation (Oebiotech, Shanghai, China). Gene expression levels were estimated from mean FPKM (fragments per kilobase of exon model per million reads mapped) values for each sample and heatmap were constructed with Z-score transforming from FPKM values using the R package ( https://www.r-project.org/, accessed on 30 March 2021). Times-series gene expression analyses were conducted by STEM (Short Time-series Expression Miner) software [25]. Pearson correlation coefficient between *SlCRK* genes and other genes were used to perform co-expressed analysis. The top 30 GO terms were collected to construct GO enrichment analysis of co-expressed *SlCRK* genes. These bioinformatic analyses were performed using the OECloud tools at https://cloud.oebiotech.cn, accessed on 4 April 2021.

### 2.4. Identification of SlCRK Family Members in the Tomato Genome

The latest released version of tomato genome SL4.0 and version ITAG4.0 of the annotation were used to search for *SlCRK* genes and sequence alignment (ftp://ftp.solgenomics.net/tomato_genome/assembly/build_4.00/, accessed on 2 March 2021). According to the annotation "cysteine-rich receptor-like protein kinase", the gene ID and protein sequence were collected. These sequences and homologous *CRK* genes in Arabidopsis were used as queries to perform BLASTP searches against the tomato protein sequence database with a maximum E-value of $1 \times 10^{-3}$ to find all remaining putative *SlCRK* genes. By eliminating genes that are not preserved in the lastest genome version or, more likely were considered as belonging to another protein family, the *SlCRK* gene family was constructed. Genomic sequences, transcript sequences and CDS sequences were all obtained. All *SlCRK* genes were submitted to EXPASy (https://web.expasy.org/protparam/, accessed on 4 March 2021) to calculate the number of amino acids, molecular weight and theoretical isoelectric points (pI). Subcellular localization predictions were conducted on WoLF PSORT (https://www.genscript.com/wolf-psort.html, accessed on 5 March 2021). The numbers roughly indicate the number of nearest neighbors for the query which localizes each site. The higher the scores, the more likely the localization is.

### 2.5. Phylogenetic Analysis

The full-length protein sequences of *SlCRK* genes were used for phylogenetic analysis. All the protein sequences were first aligned by ClustalW with default parameters [26] and then phylogenetic result was generated with MEGA X software with the maximum likelihood method, JTT substitution model with 1000 bootstrap replicates [27]. Genes were mapped on chromosomes by identifying their chromosomal position in the tomato genome database and visualized by mapgene2chromosome v2 (http://mg2c.iask.in/mg2c_v2.0/).

### 2.6. Structural Characterization and Promoter Analysis

The exon–intron organization of 35 *SlCRK* genes of tomato was analyzed using Gene Structure Display Server (GSDS) (http://gsds.cbi.pku.edu.cn/index.php, accessed on 10 March 2021) [28]. These conserved motifs of *SlCRK* family members were detected by Multiple Em for Morif Elicitation (MEME) suite 5.3.3 (https://meme-suite.org/meme/tools/meme, accessed on 10 March 2021) [29]. Conserved motifs with an e-value of $<1 \times 10^{-20}$ were subjected to further analysis. These motifs were analyzed on Pfam 34.0 database [30]. The promoter region, containing 2000 bp upstream sequences of *SlCRK* coding sequences in tomato, were retrieved from the genome sequence and then performed using PlantCARE [31] to identify the cis-regulatory elements. The cis-regulatory elements were represented by TBtools [32].

## 3. Results

### 3.1. Identification, Phylogenetic Analysis and Chromosomal Localization of SlCRK Genes in Tomato

As the annotation information of tomato genome is relatively complete, we used "cysteine-rich receptor-like protein kinase" as key words to search for target genes, and these target genes and homologous *CRK* genes in Arabidopsis were used as queries for BLASTP to search for other putative *SlCRK* genes. A total of 35 genes were identified as *SlCRK* family members. The *SlCRK* family members were named according to their gene IDs, starting from chromosome one to twelve, and their gene coordinates. The length and molecular weight of SlCRK proteins vary from 93 to 856 amino acids and 10.59 kDa to 96.25 kDa, respectively. The theoretical isoelectric point (pI) of 18 SlCRKs was acidic (4.86–6.81), and 17 CRK proteins were alkaline (7.02–9.57). According to the predicted scores, the higher of which implies a greater possibility for localization, the subcellular location of 13 SlCRKs were in the plasma membrane. Chloroplast, cytoplasm, plasma membrane and nucleus were also highly predicted location organelles. These results above were listed in Table 1. These diverse localizations of SlCRKs may indicate their various functions.

An unrooted phylogenic tree was constructed using the maximum likelihood method (Figure 1). The distribution and density of *SlCRK* genes were not uniform on chromosomes. Chromosomal localization of *SlCRK* genes are displayed in Figure 2, which shows that most *SlCRK* genes were tandemly distributed. These genes were mapped to 7 of 12 chromosomes. There are 4 members on ch01, 14 members on ch02, 6 members on ch03, 1 member on ch04, 6 members on ch05, 2 members on ch11 and ch12, respectively. Combined with phylogenic analysis and chromosomal localization, tandemly distributed *SlCRK* genes did not show high sequence similarity, suggesting they could have different functions, although they are physically related.

**Table 1.** *SlCRK* gene family members in Solanum lycopersicum.

| Gene Name | Gene ID | Gene Length, bp | Transcript Length, bp | CDS Length, bp | Protein Length, aa | pI | MW, kDa | Subcellular Location |
|---|---|---|---|---|---|---|---|---|
| *SlCRK1* | Solyc01g007960 | 2678 | 2002 | 1956 | 651 | 7.49 | 72.17 | plas: 8, E.R.: 3, nucl: 1, extr: 1 |
| *SlCRK2* | Solyc01g007970 | 3469 | 1968 | 1968 | 655 | 8.42 | 72.73 | plas: 9.5, cyto_plas: 5.5, E.R.: 4 |
| *SlCRK3* | Solyc01g007980 | 2798 | 2108 | 1917 | 638 | 8.59 | 71.30 | plas: 9, E.R.: 2, nucl: 1, extr: 1 |
| *SlCRK4* | Solyc01g007990 | 3007 | 2235 | 1917 | 638 | 8.05 | 70.72 | plas: 9.5, golg_plas: 6, vacu: 3 |
| *SlCRK5* | Solyc02g067770 | 1478 | 705 | 414 | 137 | 4.88 | 15.40 | extr: 5, vacu: 5, E.R.: 3 |
| *SlCRK6* | Solyc02g067780 | 2538 | 1652 | 1488 | 495 | 6.07 | 55.96 | chlo: 7, nucl: 3, cyto: 2, plas: 1 |
| *SlCRK7* | Solyc02g079990 | 3067 | 2242 | 2001 | 666 | 6.52 | 74.19 | plas: 10.5, golg_plas: 6.5, vacu: 2 |
| *SlCRK8* | Solyc02g080010 | 4369 | 2654 | 2571 | 856 | 7.02 | 96.25 | plas: 8.5, golg_plas: 5.5, vacu: 3, golg: 1.5 |
| *SlCRK9* | Solyc02g080020 | 624 | 504 | 504 | 167 | 8.56 | 18.41 | chlo: 4, E.R.: 4, nucl: 2, mito: 2, cyto: 1 |
| *SlCRK10* | Solyc02g080030 | 2272 | 1203 | 1203 | 400 | 5.29 | 45.05 | vacu: 9, cyto: 1, plas: 1, extr: 1, E.R.: 1 |
| *SlCRK11* | Solyc02g080040 | 2630 | 1629 | 1338 | 445 | 8.16 | 49.77 | chlo: 3, nucl: 3, cyto: 3, plas: 2.5, golg_plas: 2.5, golg: 1.5 |
| *SlCRK12* | Solyc02g080050 | 462 | 462 | 462 | 153 | 4.86 | 17.35 | chlo: 8, nucl: 2, extr: 2, cyto: 1 |
| *SlCRK13* | Solyc02g080060 | 1156 | 1104 | 1104 | 367 | 8.81 | 41.31 | vacu: 5, extr: 4, golg: 2, nucl: 1, cyto: 1 |
| *SlCRK14* | Solyc02g080070 | 4178 | 2364 | 2031 | 676 | 6.16 | 75.45 | plas: 8.5, golg_plas: 6.5, golg: 3.5 |
| *SlCRK15* | Solyc02g080080 | 3451 | 2139 | 2016 | 671 | 6.02 | 74.63 | plas: 10.5, golg_plas: 6.5, golg: 1.5 |
| *SlCRK16* | Solyc02g080090 | 2985 | 1176 | 1176 | 391 | 6.27 | 43.28 | chlo: 6, extr: 5, cyto: 2 |
| *SlCRK17* | Solyc02g082310 | 2197 | 1185 | 1185 | 394 | 4.95 | 42.37 | nucl: 7, cyto: 5, chlo: 1 |
| *SlCRK18* | Solyc02g086590 | 3939 | 2496 | 1995 | 664 | 8.56 | 72.79 | plas: 3, mito: 2, vacu: 2, E.R.: 2, golg: 2, chlo: 1, cyto: 1 |
| *SlCRK19* | Solyc03g031540 | 2260 | 681 | 681 | 227 | 6.45 | 25.56 | cyto: 8, nucl: 3, mito: 2 |
| *SlCRK20* | Solyc03g031550 | 453 | 453 | 453 | 150 | 6.11 | 17.16 | chlo: 4, extr: 4, vacu: 2.5, E.R._vacu: 2.5, mito: 2 |
| *SlCRK21* | Solyc03g111530 | 2698 | 981 | 966 | 321 | 7.42 | 36.75 | plas: 7.5, golg_plas: 6, golg: 3.5, E.R.: 2 |
| *SlCRK22* | Solyc03g111540 | 2863 | 2028 | 2028 | 675 | 6.17 | 75.90 | plas: 8.5, golg_plas: 6, vacu: 3, golg: 2.5 |
| *SlCRK23* | Solyc03g112730 | 2717 | 2230 | 1851 | 616 | 6.00 | 67.94 | E.R.: 3, chlo: 2, nucl: 2, plas: 2, extr: 2, cyto: 1, mito: 1 |
| *SlCRK24* | Solyc03g119340 | 3129 | 1676 | 1188 | 395 | 6.35 | 43.69 | nucl: 8, cyto: 3, chlo: 1, mito: 1 |
| *SlCRK25* | Solyc04g007880 | 4040 | 1468 | 1200 | 399 | 9.57 | 44.41 | nucl: 7, cyto: 3, chlo: 1, mito: 1, pero: 1 |
| *SlCRK26* | Solyc05g005050 | 1962 | 967 | 900 | 299 | 8.80 | 32.85 | vacu: 4, chlo: 3, plas: 3, extr: 3 |
| *SlCRK27* | Solyc05g005060 | 1534 | 1209 | 990 | 329 | 7.13 | 37.11 | mito: 11.5, cyto_mito: 6.5, chlo: 1 |
| *SlCRK28* | Solyc05g005070 | 3802 | 2434 | 2160 | 719 | 6.81 | 80.58 | plas: 9, nucl: 2, vacu: 2 |
| *SlCRK29* | Solyc05g018910 | 1543 | 750 | 750 | 249 | 6.31 | 27.44 | cyto: 5, golg: 3, nucl: 2, E.R.: 2, chlo: 1 |
| *SlCRK30* | Solyc05g018920 | 348 | 282 | 282 | 93 | 8.86 | 10.59 | chlo: 9, cyto: 2, vacu: 2 |

**Table 1.** *Cont.*

| Gene Name | Gene ID | Gene Length, bp | Transcript Length, bp | CDS Length, bp | Protein Length, aa | pI | MW, kDa | Subcellular Location |
|---|---|---|---|---|---|---|---|---|
| *SlCRK31* | Solyc05g018930 | 613 | 555 | 555 | 184 | 5.29 | 20.24 | chlo: 11, nucl: 1, mito: 1 |
| *SlCRK32* | Solyc11g011870 | 2769 | 1368 | 1368 | 455 | 7.97 | 50.43 | cyto: 5.5, cyto_nucl: 3.5, plas: 3, mito: 2, chlo: 1, E.R.: 1 |
| *SlCRK33* | Solyc11g011880 | 3209 | 1929 | 1929 | 642 | 8.36 | 71.20 | plas: 5.5, E.R.: 4, golg_plas: 4, vacu: 2, golg: 1.5 |
| *SlCRK34* | Solyc12g087910 | 723 | 507 | 507 | 168 | 8.85 | 19.27 | cyto: 8, extr: 3, nucl: 2 |
| *SlCRK35* | Solyc12g087920 | 3352 | 1755 | 1755 | 584 | 6.29 | 66.27 | chlo: 4, extr: 4, cyto: 2, vacu: 2, plas: 1 |

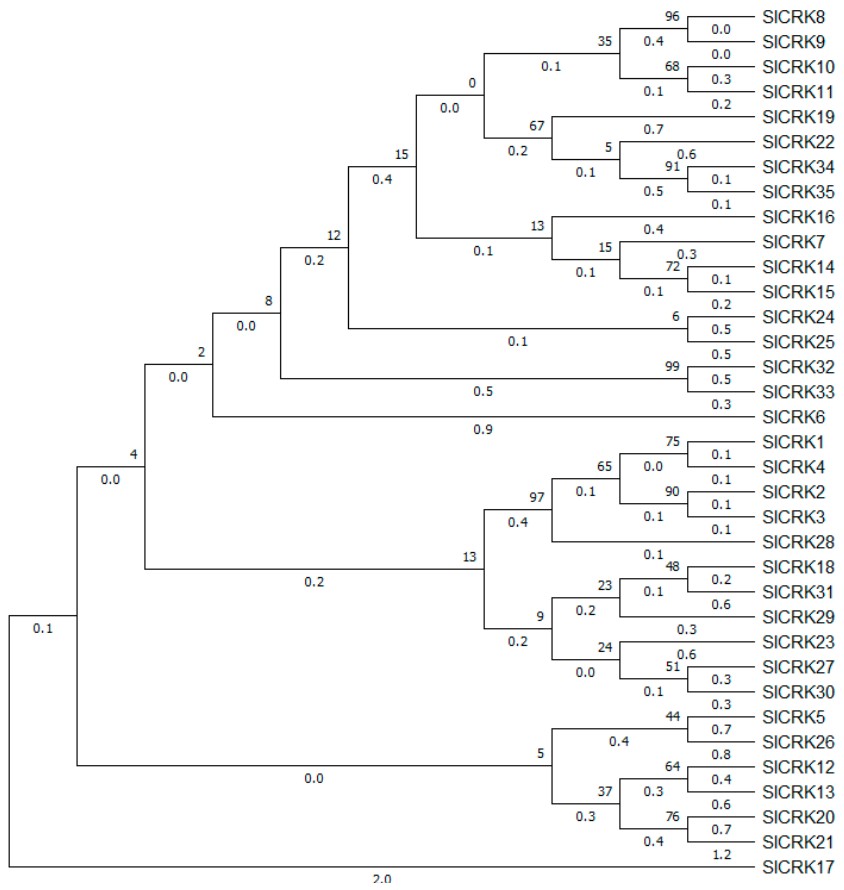

**Figure 1.** Phylogenic tree of *SlCRK* genes constructed by maximum likelihood method, JTT substitution model with 1000 bootstrap replicates. The numbers around the nodes indicate the branch lengths and the numbers below the branch indicates the frequency.

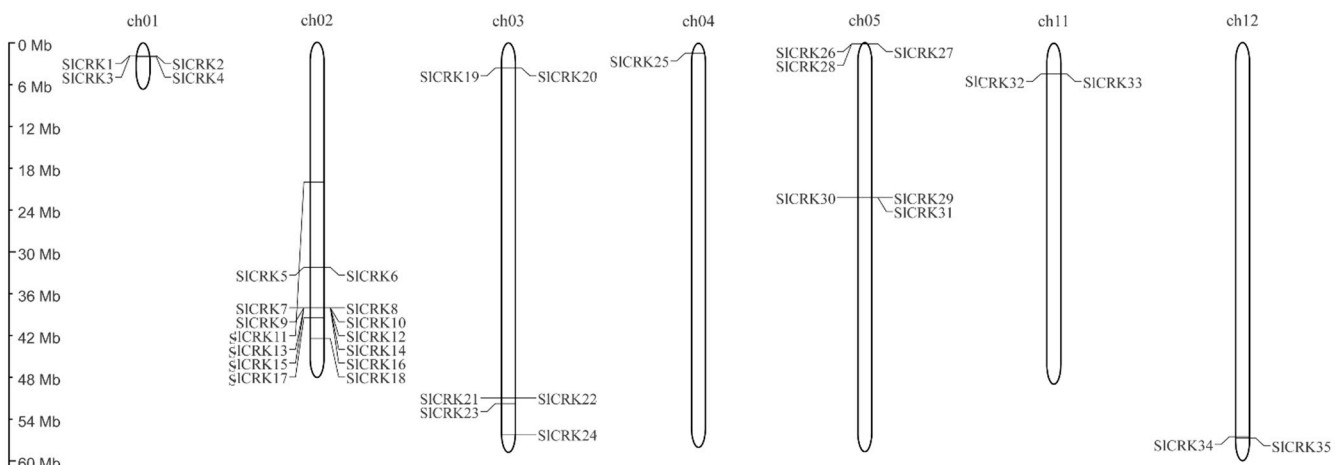

**Figure 2.** Genetic map of *SlCRK* genes in 7 chromosomes out of the 12 chromosomes of *Solanum lycopersicum*.

*3.2. Analysis of Cis-Regulatory Elements in SlCRK Genes*

To further investigate the potential regulatory mechanisms of SlCRKs during abiotic stress responses, the 2 kb upstream sequences from the translation start sites of *SlCRKs* were used to identify the cis-regulatory elements (Figure 3). In addition to missing sequence information on the promoter regions of some *SlCRKs*, the results showed that they had common upstream promoter elements, including MYB (myeloblastosis related proteins) and MYC (myelocytomatosis related proteins) binding sites, heat shock elements (HSEs)

and various responsive elements, including defense and stress responsive, light responsive, drought-inducibility, low-temperature responsive and phytohormone responsive elements. There are 359 light responsive elements and 116 HSEs in *SlCRK* promoter regions. MYB and MYC transcription factors are always associated with plant stress tolerance. There are 209 and 127 binding sites of MYB and MYC proteins in *SlCRK* upstream sequences, respectively. Besides, 61 abscisic acids responsive elements, 82 JA-responsive and 18 SA-responsive elements were predicted in *SlCRK* upstream sequences. These results indicated that *SlCRK* genes may be activated by various developmental and environmental stimuli, and *SlCRK* genes may be involved in many complex regulatory pathways.

### 3.3. Structure Analysis of SlCRKs

To study the gene structure of *SlCRKs*, we analyzed exon–intron organization and conserved motifs. The exon–intron organization analysis using a GSDS database indicated that the number and distribution of exon–intron locations were highly conserved among *SlCRK* homologs in tomato (Figure 4). *SlCRKs* have one to nine exons per gene. *SlCRK2* and *SlCRK28* have nine exons; *SlCRK5*, *SlCRK12* and *SlCRK20* have only one exon. Nine genes have seven exons which show similar distribution and exon length for each exon. Besides, phylogenetic clustered *SlCRKs* showed similar exon–intron organization while tandemly distributed *SlCRKs* did not, which is consistent with the results of the phylogenic analysis.

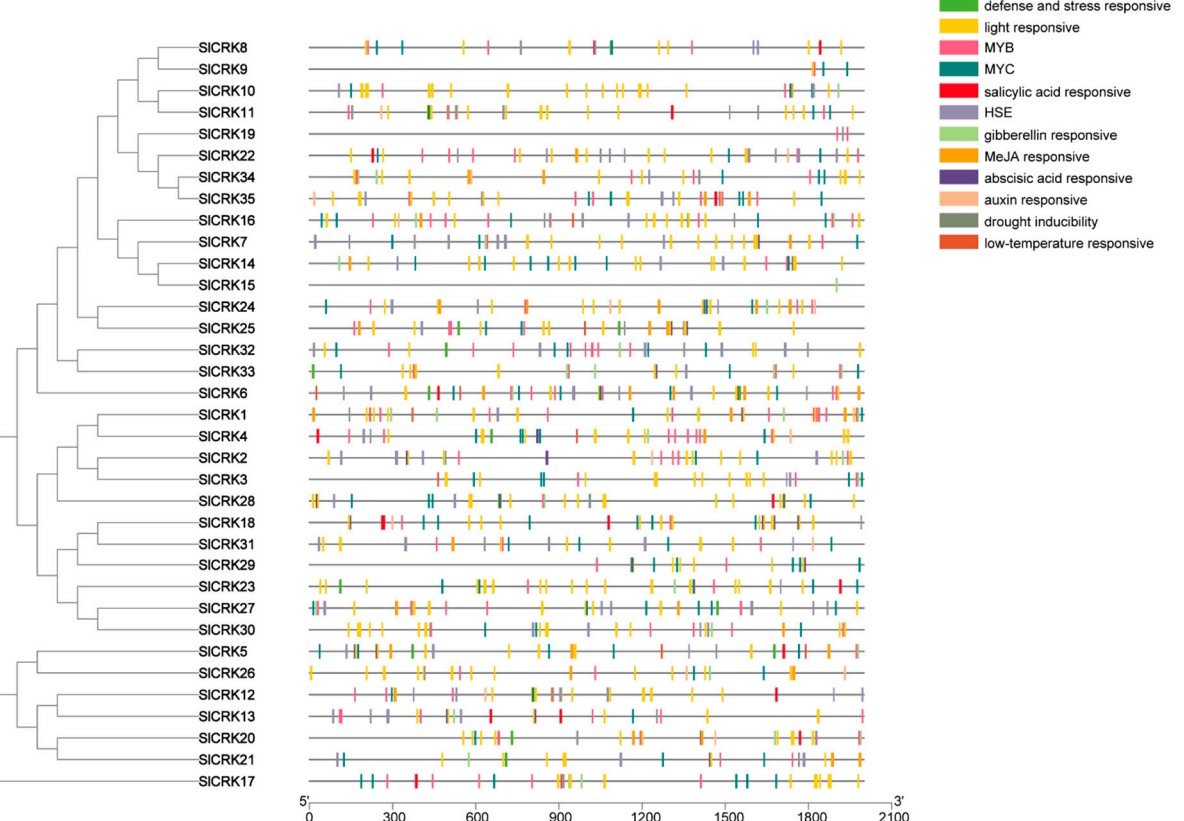

**Figure 3.** Cis-regulatory elements in *SlCRK* gene promoter regions, containing the upstream 2 kb region of *SlCRK* coding sequence.

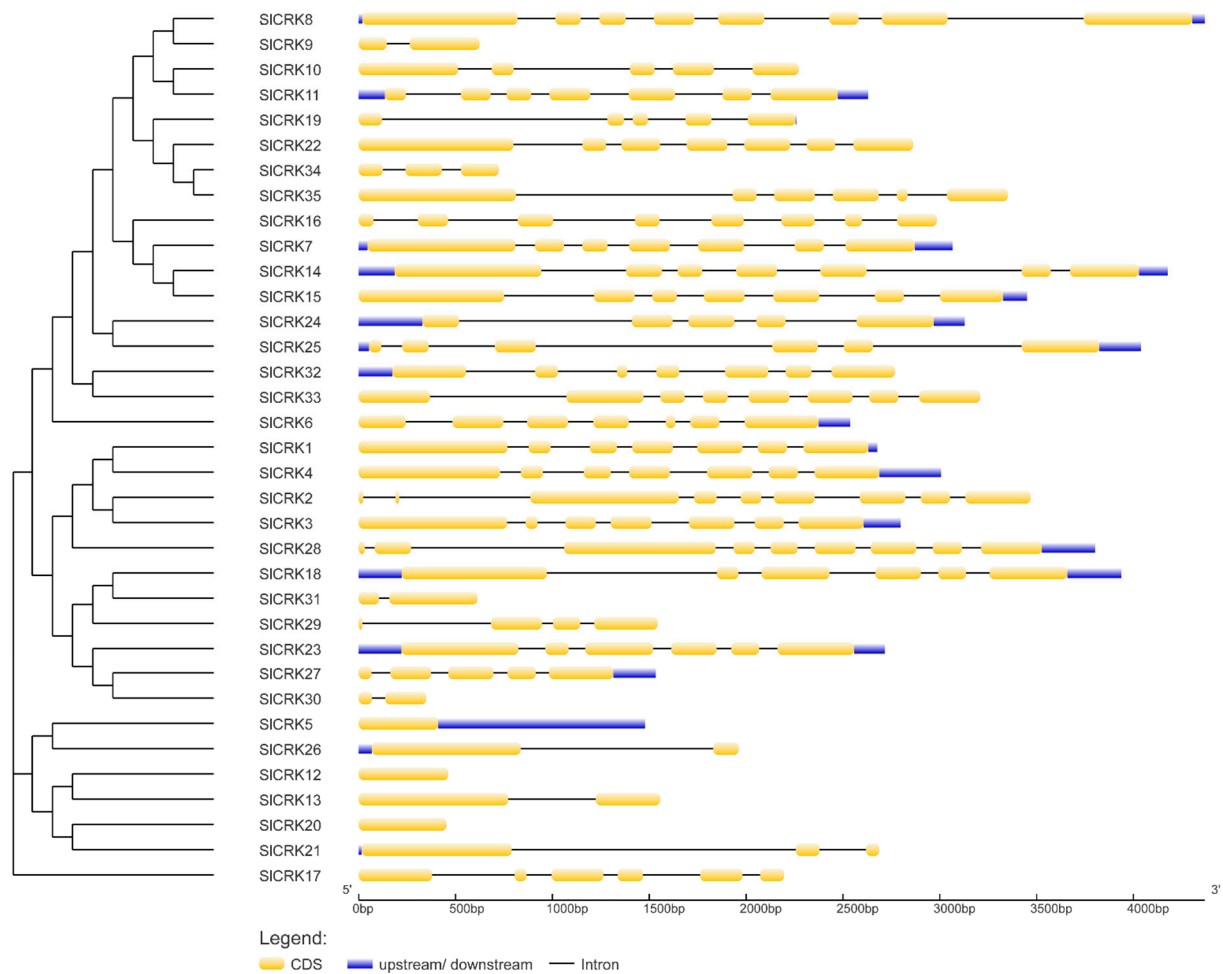

**Figure 4.** Gene structure analysis of *SlCRK* genes. Yellow boxes, blue boxes and lines indicate exons; untranslated regions indicate introns.

We then searched for conserved motifs through the MEME suite (Figure 5). SlCRKs have common conserved motifs, including salt stress response/anti-fungal domain (PF01657), protein kinase domains (PF00069) and protein tyrosine kinase (PF07714). Twenty-five of thirty-five SlCRKs have salt stress response/anti-fungal domain (PF01657); among them, 18 SlCRKs have two salt stress response/anti-fungal domains (PF01657). Twenty-four SlCRKs have tyrosine kinase domain (PF07714) and 26 SlCRKs have protein kinase domain (PF00069). SlCRK30 does not contain one of these characteristic conserved motifs from MEME, but its protein sequence is homologous to cysteine-rich receptor-like protein kinase (PTHR27002: SF128) through PANTHER (Protein ANalysis THrough Evolutionary Relationships) classification.

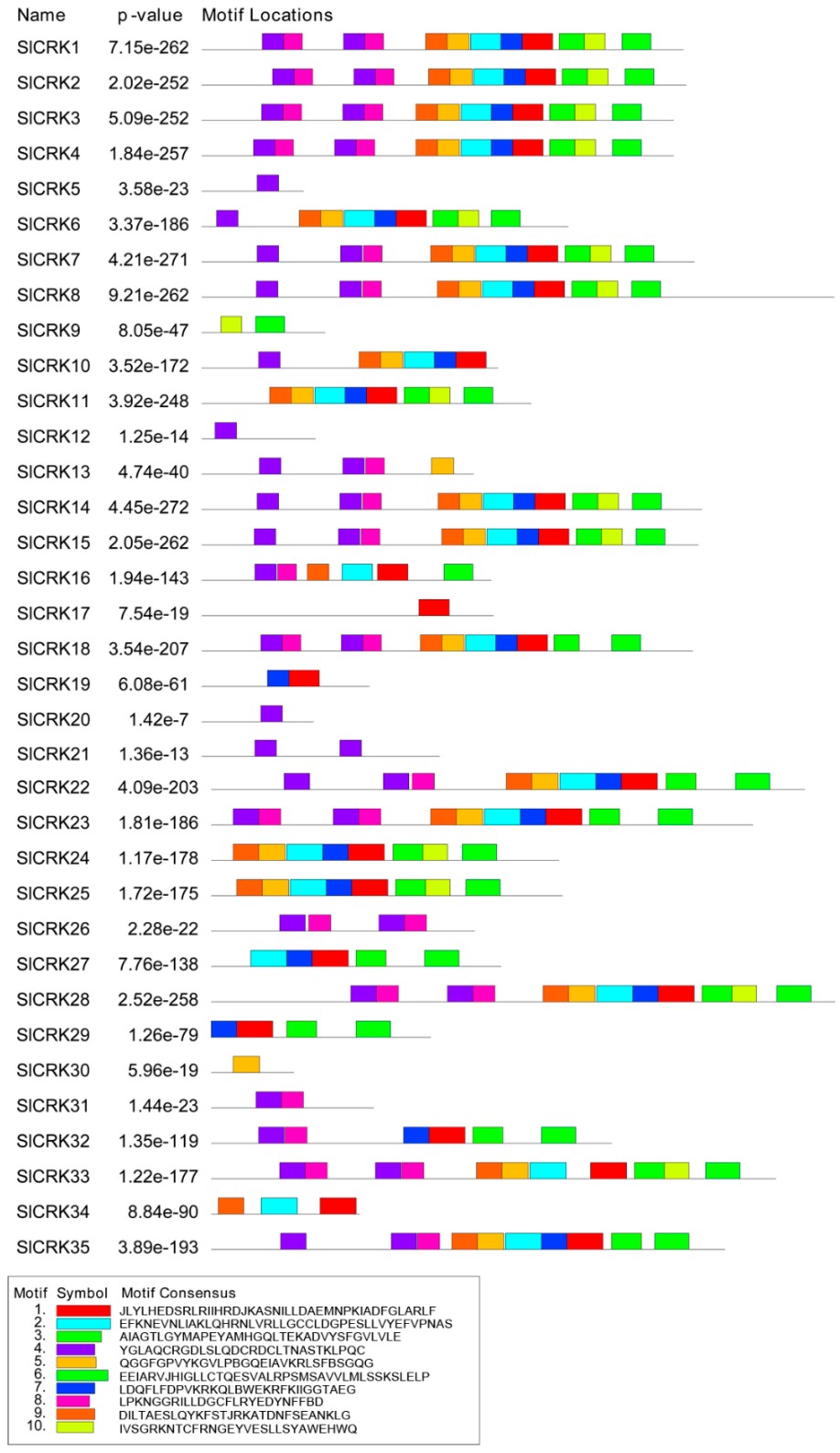

**Figure 5.** Identification of conserved motifs in SlCRK protein sequences. Significantly overrepresented motifs are graphically depicted by bars corresponding to their predicted positions. The red boxes (Motif 1) represent kinase domains (PF00069). The sapphire and grass green boxes (Motif 2 and 3) represent protein tyrosine kinase domain (PF07714) and the purple boxes (Motif 4) represent salt stress response/anti-fungal domain (PF01657). The remaining 6 motifs showed no match in the Pfam IDs.

*3.4. Expression Patterns of SlCRKs in Response to Heat Stress*

To investigate the transcript levels of *SlCRK*s in response to heat stress, we analyzed mRNA-seq results after high temperature treatment at 0 h, 24 h, 48 h and 96 h. Despite some *SlCRK*s transcripts not being detected, including *SlCRK19*, *SlCRK20* and *SlCRK29*, most *SlCRK*s were significantly down-regulated under high temperature treatment. The expression profiles were displayed as a heat map (Figure 6A). Two significant expression models were identified through short time-series expression miner (STEM) software (Figure 6B,C). Compared with expression level at 0 h, eight *SlCRK*s (*SlCRK3*, *SlCRK8*, *SlCRK9*, *SlCRK11*, *SlCRK12*, *SlCRK14*, *SlCRK15* and *SlCRK 25*) were down-regulated by two fold at 24 h and 48 h, and further by three fold at 96 h after high temperature treatment. Four *SlCRK*s (*SlCRK1*, *SlCRK7*, *SlCRK13* and *SlCRK35*) exhibited one fold down-regulated expression change at 24 h and 48 h, then decreased to three fold down-regulated at 96 h. *SlCRK2*, *SlCRK4* and *SlCRK7* showed no significant expression change at 24 h and 48 h but 1.68-fold, 1.70-fold and 1.53-fold down-regulated at 96 h. *SlCRK28* was steadily two-fold down-regulated. Adversely, *SlCRK31* exhibited close expression levels at 24 h and 96 h along with that at 0 h, but decreased to 1.83-fold at 48 h. Only two *SlCRK*s were up-regulated in response to heat stress. *SlCRK17* were up-regulated at 48 h and 96 h, and *SlCRK32* were up-regulated at 24 h. Therefore, plants adjust the expression levels of *SlCRK*s to cope with heat stress. Many *SlCRK*s were down-regulated in response to high temperature and their expression levels reached their lowest point at 96 h, especially tandemly distributed *SlCRK*s on chr02.

*3.5. Enrichment of Co-Expressed Genes Associated with SlCRKs in Tomato under High Temperature Treatment*

To further investigate the regulatory network of *SlCRK*s, we screened the genes co-expressed with *SlCRK*s and performed GO enrichment analyses. By calculating the Pearson correlation coefficient between all the mRNAs and *SlCRK* genes, a total of 6337 genes were identified, of which mRNAs with correlation coefficient greater than 0.9 and $p < 0.05$ are considered as co-expressed genes with *SlCRK*s. The top 30 GO enrichment results were shown in Figure 7A. There were several GO terms related to disease resistance, such as response to chitin (GO: 0010200), regulation of salicylic acid biosynthetic process (GO: 0080142), response to wounding (GO: 0009611), defense response (GO: 0006952), defense response to bacterium (GO: 0042742) and response to molecule of bacterial origin (GO: 0002237). Besides, phytohormone-related GO terms, including ethylene-activated signaling pathway (GO:0009873) and regulation of salicylic acid biosynthetic process (GO:0080142), and proteasomal protein catabolic process (GO:0043161, 0010498, 0010499) GO terms, were identified. Besides, calmodulin binding (GO:0005516) related genes were also co-expressed with *SlCRK*s. In addition, we screened *SlCRK* genes with significant expression model. GO enrichment of co-expressed *SlCRK* genes in model 1 was similar to that of all *SlCRK* genes (Figure 7B). Surprisingly, GO enrichment of co-expressed *SlCRK* genes in model 2 was centered on chloroplast and phtosynthesis (Figure 7C). Therefore, these results showed that there were various signaling related genes co-expressed with *SlCRK*s.

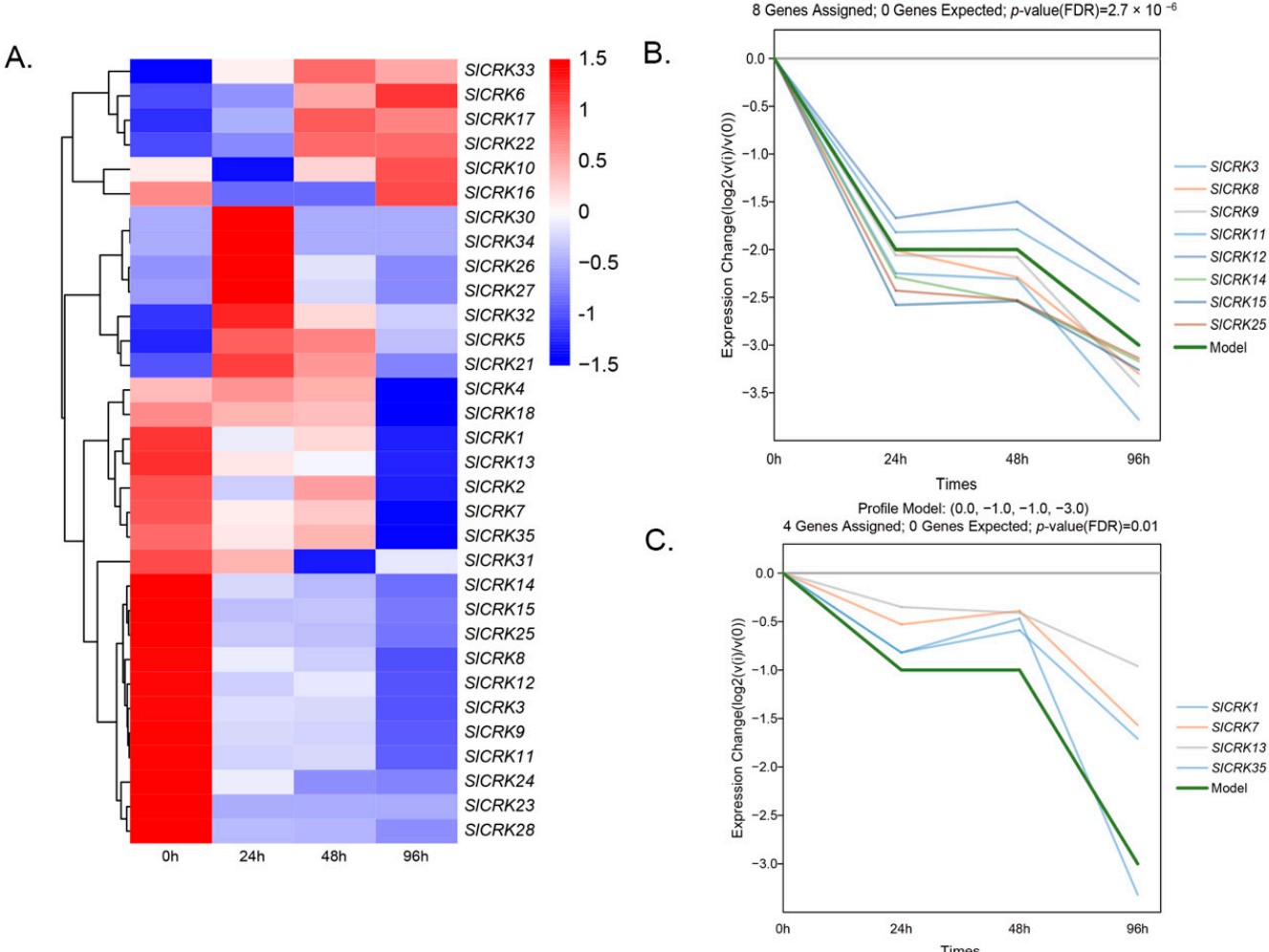

**Figure 6.** Expression pattern of *SlCRK* genes. (**A**) Heat map expression profiles of *SlCRK* genes in high temperature treatment at 0 h, 24 h, 48 h, and 96 h. The scores are Z-score transforming FPKM values. (**B**,**C**) Significant expression model $(0, -2, -2, -3)$ and $(0, -1, -1, -3)$.

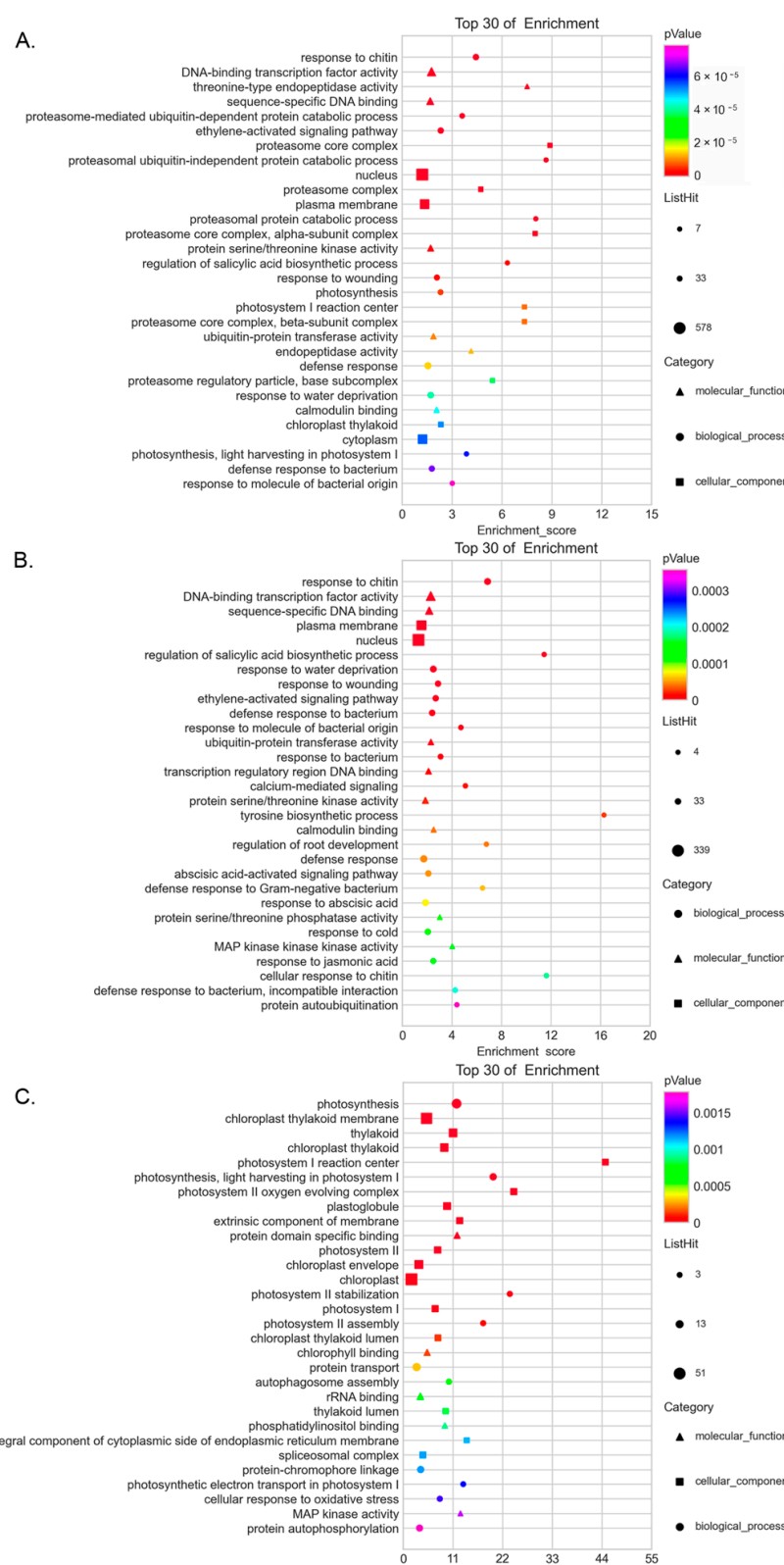

**Figure 7.** GO enrichment analysis of co-expressed genes with *SlCRK* genes. The Y axis corresponds to the GO term. The X axis corresponds to the Enrichment Score. Different shapes correspond to different GO categories (BP, CC, MF). The size of the dot corresponds to the number of different genes in the GO term and the bubble color is changed from purple-blue-green-red. The smaller the enrichment pValue, the greater the significance. GO enrichment analysis of co-expressed genes with all *SlCRK* genes (**A**), *SlCRK* genes of model 1 (**B**) and model 2 (**C**).

## 4. Discussion

RLKs in plants are a large superfamily of proteins that are structurally similar. RLKs are involved in development, growth, hormone perception and the response to pathogens [33]. CRKs are one of the largest families of RLKs, which are pivotal players in plant defense response, immunity, oxidative stress, ultraviolet radiation response, abscisic acid sensitivity and drought tolerance [5–7,10,13,15,34–37]. Their roles have been functionally uncovered in *Arabidopsis*, rice, wheat, cucumber, soybean and apple [16,17,37–40]. In *Arabidopsis*, CRK5 functions as a regulator of growth, development and ultraviolet radiation responses, enhances abscisic acid sensitivity and confers drought tolerance [35,38]. Overexpression of *CRK13* results in enhanced resistance to *Pseudomonas syringae* [7]. The semi-dominant mutation in the cysteine-rich receptor-like kinase gene, *ALS1*, confers a constitutive defense response in rice [41]. TaCRK2 contributes to leaf rust resistance in wheat [16]. Genome-wide analysis of the apple CRK family have illustrated their annotation, genomic organization, and expression profiles in response to fungal infection [38]. However, the function of CRKs in response to abiotic stress, especially heat stress, have not been investigated. Moreover, its roles in tomato have not been reported yet. Hence, to understand the putative role of *SlCRK*s under heat stress, we conducted the genome-wide identification and expression profiling of the *SlCRK* gene family in tomato.

In this study, we identified 35 *SlCRK* family members in tomato. As in *Arabidopsis*, common bean and apple [38], *SlCRK* genes were tandemly distributed in the genome of tomato (Figure 2). There are 10 *SlCRK* genes tandemly clustered on ch02. *SlCRK*s on ch01, ch03, ch05, ch11 and ch12 are also clustered, suggesting gene duplications. Only *SlCRK*17, *SlCRK*18, *SlCRK*23, *SlCRK*24 and *SlCRK*25 are separately distributed. Gene duplications, such as tandem, whole-genome, segmental duplications, are believed to have a significant impact on the expansion of gene family in flowering plants [42]. Tandem and whole-genome duplication (WGD)/segmental duplication are very common and function as a main resource of gene family expansion and genome complexity [43]. Gene clusters generally promote the recombination and accelerate evolution of the associated traits. Phylogenetic analysis (Figure 1) showed that tandemly clustered genes were not necessarily evolutionarily conserved in protein sequences, which suggested that clustered genes may not be redundant and have diverse biological functions.

Physical and biochemical property analyses were conducted on SlCRK proteins (Table 1). The length and molecular weight of SlCRK proteins vary and the theoretical isoelectric points (pI) of 18 SlCRKs were acidic and 17 SlCRKs were alkaline. The subcellular localization of 13 SlCRKs were predicted in the plasma membrane. Some were predicted in the chloroplast, nucleus, etc. The scores were listed in Table 1. These diverse locations of SlCRKs may indicate their various functions. Typical RLKs contain an extracellular ligand-binding domain, a transmembrane domain and an intracellular protein kinase domain. These results indicated that SlCRKs may have various biological functions thanks to their physically different properties.

Structure analysis and conserved motifs of SlCRKs showed many SlCRKs exhibit conserved structure and have common conserved motifs. Phylogenetic clustered *SlCRK*s, instead of tandemly clustered *SlCRK*s, showed similar exon–intron organization (Figure 4). Consistent with previous study in *Arabidopsis* and common bean, SlCRKs typically contain protein kinase domain (PF00069), tyrosine kinase domain (PF07714) and salt stress response/anti-fungal domain (PF01657) (Figure 5).

The upstream cis-regulatory elements of *SlCRK*s showed that they could be triggered by various environmental stimuli (Figure 3). The heat shock transcription factor (HSF) binds to HSEs to regulate the transcriptional response of HSF target genes [44]. There were 116 HSEs in *SlCRK* promoter regions. Plant hormone, including auxin, abscisic acid, gibberellin, Methyl jasmonate, play crucial roles in thermotolerance. To adapt to high temperature, seedlings may elevate the photosynthetic and meristematic tissues away from heat-adsorbing soil by elongating hypocotyls and therefore taking advantage of the cooling effect of moving air [45]. By blocking the auxin transport using the inhibitor of

polar auxin transport 1-aphthylphthalamic acid, the high-temperature-induced hypocotyl elongation response is abolished, indicating that auxin transport is essential to a response to increased temperature [46]. Suppressed biosynthesis of GAs interferes high-temperature-induced hypocotyl elongation in *Arabidopsis* [47]. This implies that rapid modulation of GA pathway is essential for high-temperature-induced hypocotyl elongation along with auxin. SA-dependent signaling improves basal thermotolerance but is not required for acquired thermotolerance in *Arabidopsis* [48]. Exogenous application of SA increases certain HSPs gene expression and eventually promotes heat tolerance in *Arabidopsis* [49]. In tomato, ABA-deficient mutant notabilis (not) tomato genotype is sensitive to heat stress (42 °C for 24 h), as evidenced by decreased photochemical efficiency (Fv/Fm), and increased lipid peroxidation compared with wild-type Ailsa Craig [50]. In *Arabidopsis*, JA has a positive regulatory role in basal thermotolerance, while ethylene shows the reverse effect [49,51]. It is reported that CRK5 in *Arabidopsis* is the regulator of UV light responses [35]. Light responsive cis-regulatory elements are another common motif encountered in the promoters of most of the *SlCRK*s. Defense and stress responsive and drought inducibility elements are also found in the promoter regions of *SlCRKs*, suggesting that the expression of *SlCRKs* may be triggered by biotic and abiotic stress. Besides, MYB and MYC transcription factors are key regulators in plant stress tolerance [52,53], whose binding sites occurred 209 and 127 times in *SlCRK* upstream sequences, respectively.

Expression profiles of *SlCRK* genes were visualized by heat map (Figure 6A) and two significant models in response to heat stress were identified, $(0, -1, -1, -3)$ and $(0, -2, -2, -3)$ (Figure 6B,C). Other *SlCRK*s' expression levels have been found to decrease at 24 h, 48 h and/or 96 h. These results revealed that *SlCRK*s down-regulate their expressions in response to heat stress, with different temporal distribution. Pearson correlation coefficient was calculated, and 6337 genes were identified to be co-expressed genes with *SlCRK*s. GO enrichment analysis (Figure 7) showed that under heat stress, genes of biotic stress response related GO terms (GO: 0010200, response to chitin; GO: 0009873, ethylene-activated signaling pathway; GO: 0009611, response to wounding; GO: 0006952, defense response; GO: 0042742, defense response to bacterium; GO: 0002237, response to molecule of bacterial origin) have related expression pattern with *SlCRK*s. These results indicated that *SlCRK*s may response to heat stress through similar pathway as biotic stress signaling. There may be a basal regulatory network between biotic and abiotic stress response mechanisms.

## 5. Conclusions

In summary, we performed a genome-wide analysis of *SlCRK* family members in tomato and 35 *SlCRK* genes were identified. Biochemical and physical properties of SlCRK proteins were characterized. *SlCRK*s are mostly tandemly clustered in tomato genome, with the maximum number on ch02. Gene clustering indicated the possibility of gene duplication as a factor of *CRK* gene family expansion. The phylogeny of *SlCRK*s showed that there are multiple clustered *SlCRK*s and tandemly clustered *SlCRK*s vary in protein sequence. SlCRKs share common conserved protein domain, such as protein kinase domain (PF00069), tyrosine kinase domain (PF07714) and salt stress response/anti-fungal domain (PF01657). Phylogenetic clustered *SlCRK*s have similar exon–intron organization and conserved motifs distribution. Promoter analysis identified several responsive cis-regulatory elements, including auxin, gibberellin, light and defense responsive elements. Further, most *SlCRK*s down-regulate their expression levels in response to heat stress. GO enrichment of co-expressed genes with *SlCRK*s showed that disease stress responsive genes have similar expression patterns with *SlCRK*s in response to heat stress. Taken together, our results provide a foundation for the functional characterization of SlCRK proteins in tomato.

**Author Contributions:** Y.L. and W.Z. analyzed the data and drew the figure. Y.L. and J.L. wrote the manuscript. Z.F. and Y.Z. prepared the mRNA-seq samples. J.L. and Y.Z. designed the experiments and modified the manuscript. All authors have read and agreed to the published version of the manuscript.

**Funding:** This work was supported by grants from the National Natural Science Foundation of China (32070564 and 31600207 to J.L.) and the Yunnan Fundamental Research Projects (grant No. 202101AW070002) and Shanghai Academy of Agricultural Sciences Research Project of Agricultural Science and Technology Innovation Support Field (Agricultural Science Innovation 2017 (B-06) program).

**Institutional Review Board Statement:** Not applicable.

**Informed Consent Statement:** Not applicable.

**Data Availability Statement:** The RNA-seq data generated in this study are available in the Gene Expression Omnibus (GEO) under accession number GSE174607.

**Acknowledgments:** We thank the great help from technians and OECloud tools from Shanghai OE Biotech. Co., Ltd. for their professional advices on bioinformatic analysis.

**Conflicts of Interest:** The authors declare no conflict of interest. The funders had no role in the design of the study; in the collection, analyses or interpretation of data; in the writing of the manuscript, or in the decision to publish the results.

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
