# Peer review of "Genome-Wide Identification and Characterization of Cysteine-Rich Receptor-Like Protein Kinase Genes in Tomato and Their Expression Profile in Response to Heat Stress"

_diversity, doi:10.3390/d13060258_

Round 1

Reviewer 1 Report

Diversity

The paper by Liu et al is focused on Cys rich receptor like protein kinases in Tomato.

It is quite a good paper and I have only few remarks.

Major remark: what is the rationale to make coo expression experiments with all SlCRKs? Would not be better to do it with only that of model 1 or of model 2?

Do you really expect all SlCRKs to be regulated similarly? It f not, what sense has a co expression analysis with all the SlCRKs,

Minor remarks:

Line 13: space after “,”

line 37: introduce the abbreviation DUF26

line 40 thiol redox regulation: do you mean by thioredoxin? It has been shown? Quote references.

Line 64; add “s” at “protein”

line 150: how many genes did you get by the simple annotation search?

Table I: I table legend, please explain the scores fr sub-cellular localization. What does the score mean? Is higher more likely, or the reverse?

For pI, are three decimals making sense?

Line 177: why “in addition”?

Line 190: replace “are” by “might be” only qPCR (expression analysis) can allow to conclude

line 204: please be more correct: Pkinase, Pkinase_Tyr: what is that? Use the same terminology as in line 253.

line 207 “does not”

line 215: “have not”

line 222 how do you explain SlCRK2 is not in the model?

Author Response

Dear Reviewer,

Thank you for your precious comments on our manuscript. We studied your comments and modified our manuscript based on your suggestions. The modifications and explanations are listed below. We wish these revisions could meet your requirement for publication. 

Best regards,

Yingying Zhang, Ph. D. 

The Horticulture Institute, Shanghai Academy of Agricultural Sciences

  1. Major remark: what is the rationale to make coo expression experiments with all SlCRKs? Would not be better to do it with only that of model 1 or of model 2?

Do you really expect all SlCRKs to be regulated similarly? It f not, what sense has a co expression analysis with all the SlCRKs,

Response to comment 1: Thank you for your suggestion. We added the enrichment of co-expressed genes associated with SlCRKs in model 1 and model 2, respectively, to investigate the functions of co-expression network. The figures are displayed in figure 7B-C and analysis are stated in “3.5 Enrichment of co-expressed genes associated with SlCRKs in tomato under high temperature treatment” section. These results showed that SlCRK genes in model 1 had similar functions of co-expressed genes with all SlCRK genes, and co-expressed genes of SlCRK genes in model 2 were related to photosynthesis and chloroplast.

We analyzed co-expressed genes with all SlCRK genes to investigate SlCRK genes and their expressional regulatory network. The co-expressed genes were produced by calculating the Pearson correlation coefficient between each member of SlCRK genes and every transcript in this mRNA-seq results. Those mRNA with correlation coefficient greater than 0.9 and p < 0.05 were considered as co-expressed genes. Therefore, the co-expressed genes of SlCRK genes in model 1 and model 2 were included in those of all SlCRK genes. As a result, in the manuscript we submitted before, we did co-expression analysis using all SlCRK genes. In this editon, we believed that co-expression analysis with all SlCRK genes was reasonable and could be clearer to divide into model 1 and model 2.

Minor remarks:

  1. Line 13: space after “,”

Response to comment 2: We deleted “,” to make the sentence consice.

  1. line 37: introduce the abbreviation DUF26

Response to comment 3: We added the complete name (DUF26, domain of unknown function 26) in line 38.

  1. line 40 thiol redox regulation: do you mean by thioredoxin? It has been shown? Quote references.

Response to comment 4: Thiol redox regulation means a network of sulfur metabolism in acclimation to abiotic stress, instead a single protein complex, thioredoxin. The quote reference shows on line 42 as [5].

  1. Line 64; add “s” at “protein”

Response to comment 5: Thank you for your suggestion. We corrected this in our manuscript.

  1. line 150: how many genes did you get by the simple annotation search?

Response to comment 6: Thank you for your good question. In this edition, we described the procedure of identification of SlCRK genes. According to the annotation “cysteine-rich receptor-like protein kinase”, the gene ID and protein sequence were collected. In this step, we collected 35 SlCRK genes. These profile sequences were used as queries to perform BLASTP searches against the tomato protein sequence database with a maximum E-value of 1×10−3 to find other putative SlCRK genes. In this step, we obtained extra 9 genes. However, 3 of them were not preserved in SL4.0 version, but preserved in SL2.5 version, such as Solyc02g030410; 6 of them were collected into other protein family, including lectin protein kinase family protein, G-type lectin S-receptor-like serine-threonine-protein kinase or protein kinase, such as Solyc079680, Solyc07g063760 and etc. We carefully analyzed these genes and believed that these 9 genes could not been seen as SlCRK genes. Then we prefore BLASTP searches using CRK protein sequences in Arabidopsis to identify all the putative SlCRK genes. The putative SlCRK genes have been already listed or deleted in the last step. Therefore we believe our identification of SlCRK genes are concise, accurate and consistent with the latest version of tomato genome.

  1. Table I: I table legend, please explain the scores fr sub-cellular localization. What does the score mean? Is higher more likely, or the reverse?

Responde to comment 7: The numbers roughly indicate the number of nearest neighbors to the query which localize to each site. The higher the scores, the more likely the localization is. More information could be found in this article, Horton P, Park KJ, Obayashi T, et al. WoLF PSORT: protein localization predictor. Nucleic Acids Res. 2007;35(Web Server issue):W585-W587. doi:10.1093/nar/gkm259

  1. For pI, are three decimals making sense?

Response to comment 8: Thank you for your correction. We used results from https://www.novopro.cn/tools/protein_iep.html in last submission and corrected the results by calculating pI on https://web.expasy.org/protparam/.

  1. Line 177: why “in addition”?

Response to comment 9: This is because the promoter sequences of seven SlCRK genes are not completely sequenced, including a lot of “N”. In addition to these missing sequences, the remaing results showed that SlCRK genes had common upstream promoter elements.

  1. Line 190: replace “are” by “might be” only qPCR (expression analysis) can allow to conclude

Response to comment 10: We appreciate your suggestion. We replaced “are” by “might be”.

  1. line 204: please be more correct: Pkinase, Pkinase_Tyr: what is that? Use the same terminology as in line 253.

Response to comment 11: We thank this requirement. We use the same terminology in this editon.

  1. line 207 “does not”

13 line 215: “have not”

Response to comment 12-13: We made corresponding modifications.

14 line 222 how do you explain SlCRK2 is not in the model?

Response to comment 14: The values of log2FoldChange of SlCRK2 were -0.974429026, -0.404642383 and -1.679264556, which did not conform to the tendency of model 1 and model 2.

Reviewer 2 Report

Authors analysed genes of cysteine-rich receptor-like protein kinases (CRKs) in in tomato (SlCRK). They performed bioinformatic analyses and identified 35 SlCRK genes, described their distribution on chromosomes, exon-intron organization and analyzed the presence of cis-regulatory elements in promoters of SlCRK genes. Authors did also transcriptome analysis of tomato fruits under heat stress and GO enrichment analysis of genes that were co-expressed with SlCRK.

The manuscript is well written and present interesting  and new data because the function of CRKs in response to abiotic stress, especially heat stress, have not been investigated. Combining together bioinformatics and experiments is beneficial in contrast to many theoretical manuscripts basing only on bioinformatical predictions. This manuscript should be published after minor revision.

POINTS FOR CONSIDERATION:

  • Text Format, please check whether the text format is identical with the suggested for Diversity journal (i.e. justification of paragraphs, format of subsection titles, format of figure captions).
  • Table 1 is cited first but Figure 1 appeared in the Result section before Table 1.
  • Figure 3 should be improved, instead of grey please use black color font for text to increase visibility, similarly the color code used in legend is visible but not recognizable in specific genes, please choose different colors (maybe more intense ones?).
  • Legend of Figure 5 should be improved, motif consensus sequences are so close that it is easy to get lost in these sequences, separate them.
  • References include not italicized Latin name (for example line 383), journal names are written in full with capital letters (for example line 414), in full without capital letters (for example line 409) or abbreviated (for example line 405), unify the format accordingly to the instructions for authors.

Line 25, 46 a dot is missing at the end of a sentence,

Line 28, provide words different from these used in the title

Line 57, a space is needed after the dot

Line 130, a space is needed before the citation

Lines 147-151, repetition of information given in materials and method section, please rewrite

Line 158, instead of using “etc” list all possible localizations

Page 5, italicize the Latin name in the caption  of  Table 1

Line 179, explain abbreviations

Line 180, a space is needed

Lines 185-187, literature data which is not the result of Authors, please rewrite

Line 260, abbreviation was introduced earlier,

Line 284, “number” or “members”

Lines 299-301 are not necessary because they are rewritten (copied) form results, lines155-158

Line 320, provide citation

Line 342, “to decreased” or “to decrease”

Line 347, provide names instead of numbers of GOs

Author Response

Dear Reviewer,

Thank you for your precious comments on our manuscript. We studied your comments and modified our manuscript based on your suggestions. The modifications and explanations are listed below. We wish these revisions could meet your requirement for publication. 

Best regards,

Yingying Zhang, Ph. D. 

The Horticulture Institute, Shanghai Academy of Agricultural Sciences

  1. Text Format, please check whether the text format is identical with the suggested for Diversityjournal (i.e. justification of paragraphs, format of subsection titles, format of figure captions).

Response to comment 1: Thank you for your reminder. We double checked our manuscript in this editon.

  1. Table 1 is cited first but Figure 1 appeared in the Result section before Table 1.

Response to comment 2: We adjusted the layout and made our manuscript more coherent.

  1. Figure 3 should be improved, instead of grey please use black color font for text to increase visibility, similarly the color code used in legend is visible but not recognizable in specific genes, please choose different colors (maybe more intense ones?).c

Response to comment 3: Thank you for your suggestions. We redraw figure 3.

  1. Legend of Figure 5 should be improved, motif consensus sequences are so close that it is easy to get lost in these sequences, separate them.

Response to comment 4: We made corresponding modifications.

  1. References include not italicized Latin name (for example line 383), journal names are written in full with capital letters (for example line 414), in full without capital letters (for example line 409) or abbreviated (for example line 405), unify the format accordingly to the instructions for authors.

Response to comment 5: We modified refences to uniform the format.

  1. Line 25, 46 a dot is missing at the end of a sentence,

Response to comment 6: We added the dot in this sentence.

  1. Line 28, provide words different from these used in the title

Response to comment 7: We thank reviewer for kind suggestion. However, we believe these words are most suitable as key words.

  1. Line 57, a space is needed after the dot
  2. Line 130, a space is needed before the citation
  3. Lines 147-151, repetition of information given in materials and method section, please rewrite
  4. Line 158, instead of using “etc” list all possible localizations
  5. Page 5, italicize the Latin name in the caption  of  Table 1
  6. Line 179, explain abbreviations
  7. Line 180, a space is needed
  8. Lines 185-187, literature data which is not the result of Authors, please rewrite
  9. Line 260, abbreviation was introduced earlier,
  10. Line 284, “number” or “members”
  11. Lines 299-301 are not necessary because they are rewritten (copied) form results, lines155-158
  12. Line 320, provide citation
  13. Line 342, “to decreased” or “to decrease”
  14. Line 347, provide names instead of numbers of GOs

Response to comment 8-21: Thank you for these detailed suggestions. We corrected those in this edition.

Reviewer 3 Report

Review of the manuscript “Genome-wide identification and characterization of cysteine-rich receptor-like protein kinase genes in tomato and their expression profile in response to heat stress” submitted to the Journal of Diversity. In this work, the authors performed a quite comprehensive analysis to identify the cysteine-rich receptor-like protein kinase genes in the MicroTom tomato variant and investigate their expression level through the developmental stages under high temperature treatment. However, there are a few issues  I would like authors to clarify before accepting the manuscript for any possible publication. Here are the concerns:

  • Although investigating the expression level of the identified genes through the developmental stages of the plant under high temperature can shed light on the mechanism and properties of such genes, not having a control line and comparing the point to point expression level of these genes is quite concerning if the authors are targetting to investigate and explain the gene's activity under high-temperature treatment. The expression changes might be due to developmental stages and not necessarily the response of the genes to high temperature
  • Why none of the genes expression is validated by qRT-PCR?
  • The raw RNA-Seq data should be publicly available through one of the reference public databases to allow reproducibility of the results by readers. Please provide the accession number in the manuscript
  • It seems the authors used the ITAG gene annotation information to extract the cysteine-rich receptor-like protein kinase genes. Although the reported annotation process for the ITAG4.0 gene models is quite standard, one may use other strategies that may be more appropriate such as orthologous analysis from a model organism such as Ath to identify such genes. Please verify that the genes detected are all in tomato and no cysteine-rich receptor-like protein kinase gene is missed.

Author Response

Dear Reviewer,

Thank you for your precious comments on our manuscript. We studied your comments and modified our manuscript based on your suggestions. The modifications and explanations are listed below. We wish these revisions could meet your requirement for publication. 

Best regards,

Yingying Zhang, Ph. D. 

The Horticulture Institute, Shanghai Academy of Agricultural Sciences

  1. Although investigating the expression level of the identified genes through the developmental stages of the plant under high temperature can shed light on the mechanism and properties of such genes, not having a control line and comparing the point to point expression level of these genes is quite concerning if the authors are targetting to investigate and explain the gene's activity under high-temperature treatment. The expression changes might be due to developmental stages and not necessarily the response of the genes to high temperature

Response to comment 1: Thank you for your good question. I would like to explain these concerns from two aspects below. On the one hand, the functions of CRK genes have been reported on biotic stress response and abiotic stress response, including UV radiation, osmotic stress, drought, salt treatment, salicylic acids. There have not been no investigations on heat stress response of CRK genes. We believe our findings would benefit the research progress of thermotolerance and functions of CRK genes. On the other hand, in this study, the leaves we collected have been continuously treated under heat stress. In this stage, the fruits have been mature and already turned red, and the size would no longer be larger. The state of leaves is relatively stable. Even if the expression changes might be due to developmental stages, the heat stress would be the major cause of expression changes in this stage. Meanwhile, many reported studies had similar experimental designs. We would optimize experimental design in further investigation.

For example: Zhao Y, Tian X, Wang F, et al. Characterization of wheat MYB genes responsive to high temperatures. BMC Plant Biol. 2017;17(1):208. Published 2017 Nov 21. doi:10.1186/s12870-017-1158-4

Chen S, Li H. Heat Stress Regulates the Expression of Genes at Transcriptional and Post-Transcriptional Levels, Revealed by RNA-seq in Brachypodium distachyon. Front Plant Sci. 2017;7:2067. Published 2017 Jan 10. doi:10.3389/fpls.2016.02067

  1. Why none of the genes expression is validated by qRT-PCR?

Response to comment 2: Thank you for this suggestion. We believe that RNA-seq results would be more accurate and uniform than qRT-PCR results as they use the same plant materials but rule out human error and instrument error. In addition, many reported studies preferred RNA-seq than qPCR results.

For example: Zuo J, Wang Q, Zhu B, Luo Y, Gao L. Deciphering the roles of circRNAs on chilling injury in tomato. Biochem Biophys Res Commun. 2016 Oct 14;479(2):132-138. doi: 10.1016/j.bbrc.2016.07.032. Epub 2016 Jul 8. PMID: 27402275.

Zhou R, Xu L, Zhao L, Wang Y, Zhao T. Genome-wide identification of circRNAs involved in tomato fruit coloration. Biochem Biophys Res Commun. 2018 May 15;499(3):466-469. doi: 10.1016/j.bbrc.2018.03.167. Epub 2018 Mar 27. PMID: 29580993.

Yang X, Liu Y, Zhang H, et al. Genome-Wide Identification of Circular RNAs in Response to Low-Temperature Stress in Tomato Leaves. Front Genet. 2020;11:591806. Published 2020 Nov 5. doi:10.3389/fgene.2020.591806

  1. The raw RNA-Seq data should be publicly available through one of the reference public databases to allow reproducibility of the results by readers. Please provide the accession number in the manuscript

Response to comment 3: We provided the accession number on GEO database in this edition.

  1. It seems the authors used the ITAG gene annotation information to extract the cysteine-rich receptor-like protein kinase genes. Although the reported annotation process for the ITAG4.0 gene models is quite standard, one may use other strategies that may be more appropriate such as orthologous analysis from a model organism such as Ath to identify such genes. Please verify that the genes detected are all in tomato and no cysteine-rich receptor-like protein kinase gene is missed.

Response to comment 4: Thank you for your good question. In this edtion, we described the procedure of identification of SlCRK genes. According to the annotation “cysteine-rich receptor-like protein kinase”, the gene ID and protein sequence were collected. In this step, we collected 35 SlCRK genes. These profile sequences were used as queries to perform BLASTP searches against the tomato protein sequence database with a maximum E-value of 1×10−3 to find other putative SlCRK genes. In this step, we obtained extra 9 genes. However, 3 of them were not preserved in SL4.0 version, but preserved in SL2.5 version, such as Solyc02g030410; 6 of them were collected into other protein family, including lectin protein kinase family protein, G-type lectin S-receptor-like serine-threonine-protein kinase or protein kinase, such as Solyc079680, Solyc07g063760 and etc. We carefully analyzed these genes and believed that these 9 genes could not been seen as SlCRK genes. Then we prefore BLASTP searches using CRK protein sequences in Arabidopsis to identify all the putative SlCRK genes. The putative SlCRK genes have been already listed or deleted in the last step. Therefore we believe our identification of SlCRK genes are concise, accurate and consistent with the latest version of tomato genome.
